# Addressing imbalanced data classification with Cluster-Based Reduced Noise SMOTE

**Javad Hemmatian[1], Rassoul Hajizadeh[2]\*, Fakhroddin Nazari[3]**

**1** Amol University of Special Modern Technologies, Amol, Iran, **2** Machine Learning and Deep Learning Laboratory, Faculty of Engineering Modern Technologies, Amol University of Special Modern Technologies, Amol, Iran, **3** Faculty of Engineering Modern Technologies, Amol University of Special Modern Technologies, Amol, Iran

\* r.hajizadeh@ausmt.ac.ir

**Data Availability Statement:** All relevant data are within the paper and its Supporting information files.

**Funding:** The author(s) received no specific funding for this work.

## Abstract

In recent years, the challenge of imbalanced data has become increasingly prominent in machine learning, affecting the performance of classification algorithms. This study proposes a novel data-level oversampling method called Cluster-Based Reduced Noise SMOTE (CRN-SMOTE) to address this issue. CRN-SMOTE combines SMOTE for oversampling minority classes with a novel cluster-based noise reduction technique. In this cluster-based noise reduction approach, it is crucial that samples from each category form one or two clusters, a feature that conventional noise reduction methods do not achieve. The proposed method is evaluated on four imbalanced datasets (ILPD, QSAR, Blood, and Maternal Health Risk) using five metrics: Cohen's kappa, Matthew's correlation coefficient (MCC), F1-score, precision, and recall. Results demonstrate that CRN-SMOTE consistently outperformed the state-of-the-art Reduced Noise SMOTE (RN-SMOTE), SMOTE-Tomek Link, and SMOTE-ENN methods across all datasets, with particularly notable improvements observed in the QSAR and Maternal Health Risk datasets, indicating its effectiveness in enhancing imbalanced classification performance. Overall, the experimental findings indicate that CRN-SMOTE outperformed RN-SMOTE in 100% of the cases, achieving average improvements of 6.6% in Kappa, 4.01% in MCC, 1.87% in F1-score, 1.7% in precision, and 2.05% in recall, with setting SMOTE's neighbors' number to 5.

## 1. Introduction

Machine Learning (ML) is a subfield of artificial intelligence (AI) that has seen remarkable growth and advancement in recent years [1, 2]. The primary goal of machine learning is to enable machines to learn independently, reducing the need for human intervention [3]. ML algorithms have become integral to various industries, contributing significantly to sectors such as medicine applications, optical character recognition [4], medical image processing [5], wireless communications [6], software defect prediction [7, 8], self-driving cars [9], and image recognition [10]. For instance, the StackedEnC AOP [11] model excels in forecasting antioxidant proteins, while the iAFPs Mv BiTCN [12] model leverages self-attention transformer

**Competing interests:** The authors have declared that no competing interests exist.

embedding's for precise antifungal peptide predictions. Additionally, the AIPs DeepEnC GA [13] model combines evolutionary features with a genetic algorithm based deep ensemble approach to effectively predict anti-inflammatory peptides. These cutting edge models demonstrate the significant role of machine learning in advancing research. Deepstacked AVPs, SnTCN AIPs [14, 15] can be mentioned among the state-of-the-art models in this field. The effectiveness of these algorithms is a key driver of progress in these fields [16]. Classifications in machine learning are crucial as they allow machines to make decisions and predictions based on data.

In general, classification section is an unavoidable part of machine learning applications. One of the problems is the poor performance of classifiers in dealing with imbalanced data. In imbalanced data, the amount of data in each class is significantly different from the other classes [17]. There are inherent challenges in learning from class-imbalanced data. The skewed distribution of training examples causes standard classifiers to be biased, favoring the majority class and struggling to detect rare instances [18]. Accuracy metrics for classifiers often prove unreliable due to their failure to account for minority classes. For example, in a dataset where 90% of samples represent healthy individuals and only 10% have cancer, this imbalance can severely hinder the model's ability to accurately identify cancer cases. Consequently, the model may achieve high accuracy overall by correctly classifying the majority class (healthy individuals) while performing poorly in identifying the minority class (cancer cases). Imbalanced datasets can lead to several issues, including:

a) *Bias toward majority class*: Models may become biased toward the majority class, resulting in poor performance on the minority class.

b) *Misleading Accuracy*: High overall accuracy can be misleading, as it may reflect the model's ability to predict the majority class rather than its effectiveness in identifying the minority class.

c) *Poor Generalization*: The model may struggle to generalize to new, unseen data, particularly for the minority class.

d) *Increased False Negatives*: There is a higher likelihood of misclassifying minority class instances, leading to increased false negatives.

In the field of machine learning, data skewness has led many researchers to concentrate on class-imbalanced learning [19]. Addressing the classification problem of imbalanced data is a vital area of research in machine learning. The literature offers a range of methods, as illustrated in Fig 1, including data-level, algorithm-level, and hybrid approaches, to tackle this issue [20]. Below is an overview of key methods and strategies discussed in the literature for addressing class imbalance in machine learning.

Data-level techniques primarily focus on modifying the dataset to create a more balanced representation of classes. This can be achieved by either reducing the number of samples in the majority classes or increasing the number of samples in the minority classes [21]. Currently, data-level methods primarily focus on data preprocessing, utilizing resampling to redistribute the training data across different classes [22]. One advantage of data-level techniques is that resampling and classifier training are independent of each other [23]. Resampling methods can be classified into three types: (i) undersampling, (ii) oversampling, and (iii) combined methods [24]. Fig 2 illustrates an example of an imbalanced dataset that is balanced using two methods: undersampling and oversampling.

Oversampling involves increasing the number of instances in the minority class to match the majority class. A common technique is random oversampling, where instances from the

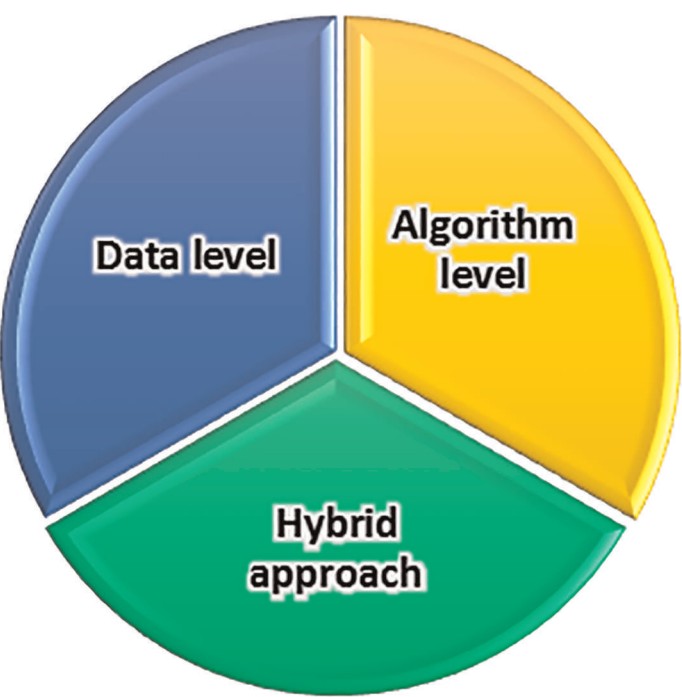

**Fig 1. Taxonomy of approaches to addressing imbalanced data issue.**

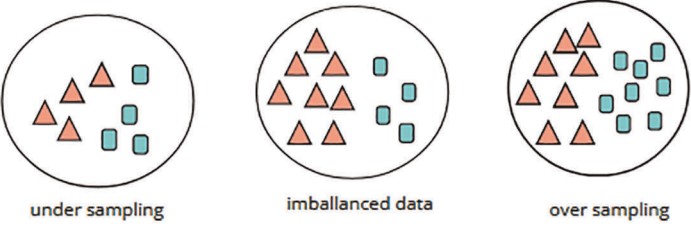

**Fig 2. Concepts of oversampling and undersampling techniques.**

minority class are duplicated. Synthetic Minority Over-sampling Technique (SMOTE) generates synthetic samples for the minority class by interpolating between existing instances [25]. This method has been widely used due to its ability to reduce overfitting while improving model performance on the minority class. Borderline-SMOTE [26] and ADASYN [27] (Adaptive Synthetic Sampling) are variants of SMOTE. They focus on generating synthetic samples near the decision boundary or difficult samples, aiming to create better class separability and improve model performance near the decision boundary.

Random undersampling reduces the size of the majority class by randomly removing samples, making the classes more balanced [28]. However, it risks losing valuable information. Cluster-based undersampling methods, such as k-means [29], are applied to the majority class to select representative samples, preserving diversity in the data while reducing the sample size.

Combining oversampling and undersampling (e.g., SMOTE with undersampling) has been shown to improve model robustness and reduce overfitting and information loss [30].

Algorithm-level methods modify the learning process or the objective function to account for imbalanced classes [31]. These techniques can be categorized into several types, including ensemble-based methods, threshold methods, one-class learning, cost-sensitive learning, and active learning methods [32]. Generally, cost-sensitive methods are more appealing to researchers compared to other available approaches [33]. Cost-sensitive modifications approaches assign higher misclassification costs to the minority class. This can be achieved by modifying the loss function, such as using Weighted Cross-Entropy Loss in neural networks or Gini Impurity for decision trees, where higher penalties are applied to misclassified minority samples [34]. Instead of altering the data or model, thresholds can be adjusted after model training to favor the minority class by shifting the classification decision boundary. Using ensemble methods is a common method in algorithm level. Boosting techniques (e.g., Ada-Boost, Balanced Random Forest) adjust sample weights or focus on misclassified instances, effectively giving more attention to the minority class [35]. Balanced Random Forests introduce class weights to ensure that both classes are equally represented during the model-building process. Variants like Easy Ensemble and Balance Cascade create balanced subsets of data by combining undersampling with bagging techniques, training separate classifiers for each balanced subset and aggregating their results [36]. Also, ensembles combining boosting and bagging techniques have shown promise in improving classification performance for minority classes.

Hybrid Approaches of overcoming imbalanced data combine data-level and algorithm-level techniques to leverage the strengths of both. The key idea is to leverage the strengths of different methods to achieve better overall performance [37]. Some studies propose the hybridization of sampling and cost-sensitive learning [38]. SMOTE with Cost-Sensitive Algorithms applies SMOTE to balance the dataset, followed by training with a cost-sensitive algorithm to further optimize performance for the minority class. Hybrid Sampling with ensemble learning is a combination of oversampling the minority class and undersampling the majority class, followed by training an ensemble of classifiers. SMOTEBoost [39] and Random Undersampling Boosting (RUSBoost) [40] are two notable examples that integrate data balancing within ensemble learning frameworks.

In addition, deep learning models have been adapted to address class imbalance by modifying architectures, loss functions, or training techniques. In neural networks, loss functions are adjusted to penalize minority class misclassification more heavily (Class-weighted Loss Functions). Weighted cross-entropy loss is commonly used, but more advanced options like Focal Loss focus on harder-to-classify samples by reducing the importance of easy examples [41]. In computer vision, augmentation techniques such as rotations, flips, and color variations can generate synthetic data for minority classes. Generative Adversarial Networks (GANs) have been used to generate new samples for minority classes, particularly in image and text data [42].

Furthermore, imbalanced data scenarios can also be treated as anomaly detection problems, where the minority class is considered an anomaly. Models like autoencoders or One-Class SVMs are trained on the majority class, and instances of the minority class are detected as outliers during testing [43].

Since traditional metrics like accuracy are misleading in imbalanced scenarios, researchers use metrics that better capture the performance on minority classes. Precision, Recall, and F1-score metrics focus on the performance specific to each class, providing insights into the true positive and false positive rates. Area Under the ROC Curve (AUC-ROC) considers the true positive rate against the false positive rate, which is helpful for evaluating models on

imbalanced data. When classes are highly imbalanced, the precision-recall curve is often a more reliable metric than AUC-ROC, as it emphasizes minority class performance [44].

Addressing class imbalance in machine learning remains a multi-faceted challenge that requires careful consideration of data characteristics, domain-specific requirements, and computational resources. The choice of method(s) often depends on the dataset, application context, and model complexity. For instance, resampling and ensemble techniques are commonly applied in traditional ML algorithms, whereas cost-sensitive learning and custom loss functions are popular in deep learning. Hybrid approaches, combining multiple methods, tend to yield better results in highly imbalanced settings, as they leverage the advantages of data-level and algorithm-level adjustments.

## 1.1 motivation and contributions

This article primarily focuses on data-level approaches. Typically, proposed data-level methods do not consider classification applications and merely aim to balance the data samples. In contrast, our proposed method addresses imbalanced data with a focus on classification applications. We introduce a three-stage oversampling method similar to the Reduced Noise Synthetic Minority Oversampling Technique (RN-SMOTE) [45], which is a state-of-the-art approach in this area. Our method involves oversampling, removing noisy samples, and then oversampling again. In the removal step, the classification concept is applied to ensure that noise reduction does not split samples of a class into more than two clusters.

The proposed Cluster-Based Reduced Noise SMOTE (CRN-SMOTE) method addresses the critical challenge of imbalanced data classification by integrating SMOTE and DBSCAN techniques. Traditional methods often neglect the clustering characteristics of minority class samples, leading to the creation of multiple clusters within a single category. CRN-SMOTE innovatively focuses on maintaining one or two concentrated clusters, which is essential for effective classification. By systematically balancing the data, identifying and removing noisy samples, and controlling the clustering process, this method enhances the quality of synthetic samples. The contribution lies in its ability to preserve the integrity of class distributions while significantly reducing noise, ultimately improving classification performance. This approach not only outperforms existing methods like RN-SMOTE but also provides a robust framework for handling imbalanced datasets across various applications. The significant contributions are outlined below:

- Integrate SMOTE with a cluster-based approach to create high-quality synthetic samples for minority classes, addressing the limitations of traditional oversampling techniques.

- Enforce a constraint on the number of clusters formed per category in noise reduction step, ensuring that samples remain concentrated in one or two clusters, which is crucial for effective classification.

- Demonstrate significant improvements in key performance metrics (Kappa, MCC, F1-score, Precision, and Recall) compared to state-of-the-art methods.

- Offer empirical evidence of its effectiveness through rigorous evaluation on multiple imbalanced datasets, showcasing its superiority in maintaining class integrity and enhancing model performance.

The rest of the manuscript is organized as follows: In Section 2, we review and evaluate related works and existing methods. Section 3 introduces the proposed method and discusses how to adjust the parameters. In Section 4, titled "Simulation Results," we present the dataset, the metrics for evaluating imbalanced data, and the simulation results, followed by a discussion

of these results. Future work is outlined in Section 5. Finally, Section 6 draws conclusions based on the findings.

## 2. Related works

In this study, we propose a novel cluster-based Reduced Noise SMOTE approach that can be integrated into state-of-the-art data-level methods to address the issue of imbalanced data, specifically the RN_SMOTE method [45]. This section reviews both SMOTE and RN_SMOTE methods, followed by an overview of Density-Based Spatial Clustering of Applications with Noise (DBSCAN), a density-based technique for noise reduction [46].

### 2.1. Synthetic Minority Oversampling Technique (SMOTE)

An effective data-level method for addressing imbalanced data is SMOTE [25]. SMOTE is an oversampling technique used to balance the original training dataset. Rather than simply repeating minority class samples, the key idea behind SMOTE is to generate synthetic samples. These synthetic samples are specifically created to mimic the characteristics of the original data.

Suppose $X$ is a data matrix with $m$ different classes. $X^{(C_i)}$ represents the matrix of data for class $C_i$, (where $i = 1, \ldots, m$), and $N_{(C_i)}$ is the number of samples in class $C_i$. In cases of imbalanced data, there is typically at least one class with significantly more samples $(N_{(C_{max})})$ compared to others.

In SMOTE, after identifying the majority class, the number of synthetic samples to be generated for each minority class is calculated. For each minority class, synthetic samples are created using the original samples and their neighbors. The $K$ (i.e., $K_{SMOTE}$) nearest neighbors $(X_{KNN_h}^{(c_i)})$ of each sample $(x_h^{(c_i)})$ are determined using the Euclidean distance. The synthetic sample $(x_{Syn_h}^{(c_i)})$ is then generated using the following equation:

$$x_{Syn_h}^{(c_i)} = x_h^{(c_i)} + \beta \times \left( rand_x^{(c_i)} - x_h^{(c_i)} \right) \tag{1}$$

where $rand_x^{(c_i)}$ is a randomly selected sample from the neighbors of $x_h^{(c_i)}$, i.e., $X_{KNN_h}^{(c_i)}$, and $\beta$ is a random value between 0 and 1. The synthetic sample is incorporated into the original samples, and this process iterates until the number of samples in the minority class equals that of the majority class. In the end, the number of samples in each class is equal to $N_{(C_{max})}$. Algorithm 1 provides a detailed overview of the upsampling process in imbalanced datasets using SMOTE.

Algorithm 1. SMOTE($K_{SMOTE}$, X).

```
Input: Imbalanced data (X), K_SMOTE (number of neighbors)
Output: Balanced data
• Determine the number of samples of the class with maximum samples
  (N_(C_max))
• For each minority class X^(c_i) with sample number of N_c_minority_i
  set the number of synthesis data to zero: C_syn_data = 0,
  while C_syn_data + N_c_minority_i < N_(C_max)
    Randomly select a sample from X^(c_i), denoted as x_h^(c_i)
    Determine the K nearest neighbors of x_h^(c_i), referred to as X_KNN_h^(c_i),
    Randomly select a sample from X_KNN_h^(c_i), denoted as rand_x^(c_i)
    Generate the synthetic sample: x_Syn_h^(c_i) = x_h^(c_i) + β × (rand_x^(c_i) − x_h^(c_i)),
    Insert the synthesis data (x_Syn_h^(c_i)) into the imbalanced data X,
    Increment the count of synthetic data: C_syn_data = C_syn_data + 1
```

This type of data oversampling is widely used in fields such as data science and machine learning. It provides a secure platform for algorithms to improve their performance without compromising privacy or the safety of real data. Additionally, it can enhance existing datasets, particularly in cases where the original data is limited or biased. SMOTE is a valuable tool for generating synthetic data and creating balance between classes in imbalanced datasets.

These new data points are created by interpolating between several minority class samples defined within a neighborhood. For this reason, the approach is said to focus on the "feature space" rather than the "data space." In other words, the algorithm considers the values and relationships of features rather than individual data points [47].

## 2.2. Density-Based Spatial Clustering of Applications with Noise (DBSCAN)

The DBSCAN algorithm, introduced by Martin Ester et al. in 1996, is a density-based clustering method capable of discovering arbitrary clusters within a dataset [48]. One of its main advantages over other clustering methods is its ability to identify clusters of various shapes without requiring prior knowledge of the number of clusters. DBSCAN is capable of identifying areas dense enough to be classified as clusters [49]. This clustering method relies on specific user-defined parameters, including the neighborhood radius ($\varepsilon$) and the minimum number of points [50] within the neighborhood. In DBSCAN, each data point can be categorized into one of the following three types: core points, noise points, and border points.

A data point is labeled as a core point when it is surrounded by a minimum number of neighboring points (including itself) within a proximity of $\varepsilon$. These core points serve as the focal points of a cluster. A data point is classified as a noise point, or outlier, when it lacks sufficient neighboring points within the specified distance ($\varepsilon$) and does not meet the criteria to be a core point. Noise points are not part of any cluster. A data point that is not a core point but lies within the $\varepsilon$ distance of a core point is referred to as a border point. Border points are on the edge of a cluster and may be considered part of the cluster or noise, depending on the clustering criteria [51]. Fig 3 illustrates the placement of all three types of data points based on a minimum of 5 samples and an $\varepsilon$ radius.

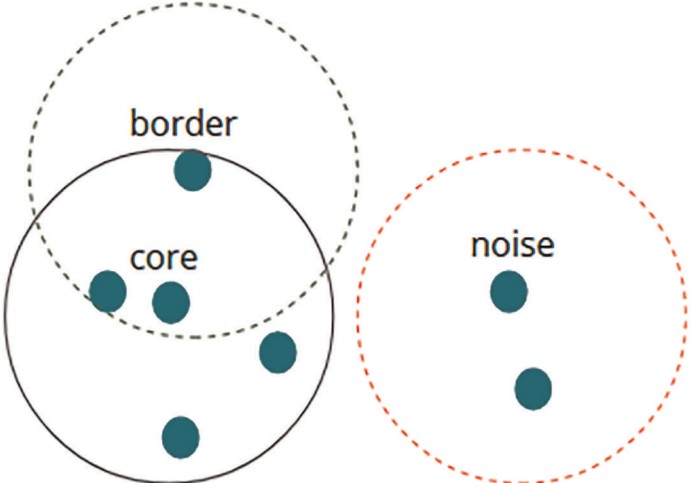

**Fig 3. An example of using DBSCAN to categorize data points of a dataset for fixed radius ($\varepsilon$) and the minimum number of samples = 5.**

The steps of DBSCAN algorithm are given in Algorithm 2. In the given DBSCAN algorithm, the border points are considered as part of the cluster. In DBSCAN, the parameters $\varepsilon$ and $N$ are usually set empirical. Also, an algorithm has been introduced which is performed corresponding to distances among samples, which is called *K-distance*. The K-distance algorithm calculates the distances between a point and its K-nearest neighbors in the dataset. These distances are used to determine the density reachability and core distance, which are essential for identifying clusters and outliers in DBSCAN. DBSCAN

Algorithm 2. DBSCAN($\varepsilon$, N).

```
Input: input data (D), neighborhood radius (ε), minimum number of sam-
       ples (N)
Output: Remove data points detected as noise and provide clustered,
        clean data.
For each sample in D, denoted as dᵢ:
  • Calculate the number of samples within the neighborhood radius ε
    of dᵢ, referred to as Nᵢ,
  • If Nᵢ ≥ N, categorize dᵢ as a core point,
  • Else:
    • If there is a core point in the neighborhood of dᵢ, categorize dᵢ
      as a border point.
    • Otherwise, categorize dᵢ as a noise point.
Remove noise points and cluster the data points that are connected to
one another.
```

## 2.3. Reduced Noise-SMOTE (RN-SMOTE)

The RN-SMOTE algorithm is a data-level oversampling method designed to address the challenges of imbalanced data. As illustrated in Fig 4, RN-SMOTE consists of three primary stages that enhance class distribution and minimize the negative impact of noise on synthetic samples.

In the initial stage, RN-SMOTE applies the SMOTE technique to balance the class distribution by generating synthetic samples for the minority class. In the second stage, the RN-SMOTE algorithm employs the DBSCAN algorithm to mitigate the noise effect on the synthetic samples generated by SMOTE. DBSCAN identifies clusters of high-density data points and separates them from low-density regions, which are often considered noise. By applying DBSCAN, RN-SMOTE aims to refine the synthetic samples and reduce the potential negative impact of noise on the model's performance. In the final stage, the RN-SMOTE algorithm ensures that the class distribution is well-balanced, without being biased towards the majority class. In this stage, SMOTE algorithm is again utilized to restore balance to the data, making it ready for classification.

This balanced class distribution contributes to improved model performance and a more accurate understanding of the underlying patterns in the data. In summary, the RN-SMOTE algorithm combines SMOTE and DBSCAN to balance class distribution and reduce noise. This approach leads to more robust and accurate models of classification when dealing with imbalanced datasets [45].

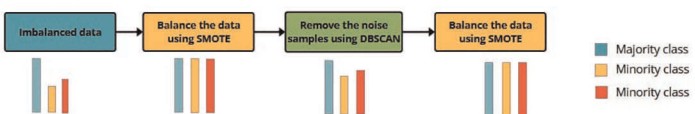

**Fig 4. The stages of RN-SMOTE.**

## 3. Proposed method: Cluster-based RN-SMOTE (CRN-SMOTE)

In this study, a cluster-based approach is presented using SMOTE and DBSCAN techniques to address imbalanced data classification. In common oversampling methods, there is not any attention to the synthetic samples of each class, and it causes samples in a category are placed in several clusters. While the samples of a category usually make one or two concentrated clusters. The proposed method, similar to RN-SMOTE, includes three stages. Firstly, the data are balanced using SMOTE technique. Then, the noisy samples are determined and removed to reduce the noise effect with a limit on the number of clusters created per category. The number of clusters is set to one or two, because the samples belong to the same category. Finally, after reduction the noise effect, the data again is balance using SMOTE. The proposed method is called Cluster-based Reduced Noise SMOTE (*CRN-SMOTE*). Algorithm of the proposed CRN-SMOTE is given in the following:

Algorithm 3. Proposed cluster based RN-SMOTE.

**Input:** imbalanced data ($\boldsymbol{X}$), $N_{cluster}$ (desired number of clusters), $N$ (minimum number of samples), $K_{SMOTE}$

**Output:** balanced data.

- Balance the samples of different categories $\boldsymbol{X}^{(C_i)}$ using SMOTE ($K_{SMOTE}$, $\boldsymbol{X}$)
- For each SMOTEed category $\boldsymbol{X}^{(C_i)}_{SMOTE}$:
  - Set $\varepsilon$ to a small positive value (e.g., $\varepsilon = 0.1$)
  - Repeat while
    Apply DBSCAN($\varepsilon$, $N$) to remove noise samples $\boldsymbol{X}^{(C_i)}_{DBSCAN}$ for each category
    Calculate the number of sample clusters in $\boldsymbol{X}^{(C_i)}_{DBSCAN}$, denoted as $N^{(C_i)}_{DBSCAN}$
    If $0 < N^{(C_i)}_{DBSCAN} \leq N_{cluster}$
      End while
    Else:
        Increase $\varepsilon$ by a small positive value: $\varepsilon + \delta \to \varepsilon$, (where $\delta$ is a small positive value)
- Concatenate the category with maximum samples with the cluster-based
  reduced noise categories: $\boldsymbol{X}_{DBSCAN} = \begin{bmatrix} \boldsymbol{X}^{(C_{max})}, \bigcup^{m}_{\substack{i=1 \\ i \neq max}} \boldsymbol{X}^{(C_i)}_{DBSCAN} \end{bmatrix}$
- balance the samples of different categories in $\boldsymbol{X}_{DBSCAN}$ using SMOTE
  ($K_{SMOTE}$, $\boldsymbol{X}_{DBSCAN}$) resulting in $\boldsymbol{X}_{CRN-SMOTE}$

In Algorithm 3, after balancing the categories $\boldsymbol{X}^{(C_i)}$ of the dataset using SMOTE, noisy samples are removed from the SMOTEed categories ($\boldsymbol{X}^{(C_i)}_{SMOTE}$), with a limit on the number of clusters for the residual samples. A modified DBSCAN method is employed to eliminate noisy samples and determine the desired number of clusters for each category. In the proposed cluster-based method, unlike the standard DBSCAN approach, there is no need to predefine the neighborhood radius ($\varepsilon$). Instead, $\varepsilon$ is set to a small positive value along with a specified minimum number of samples. This initial parameter setting for DBSCAN typically leads to the formation of clusters with a higher number of residual samples (usually more than two). Subsequently, the $\varepsilon$ value is incrementally increased by a small positive value ($\delta$) until the limitation on the number of clusters is satisfied:

$$0 < N^{(C_i)}_{DBSCAN} \leq N_{cluster}, \tag{2}$$

where $N^{(C_i)}_{DBSCAN}$ is the number of clusters in residual samples for category $C_i$, and $N_{cluster}$ is defined as the desired number of clusters.

In the proposed method, the desired number of clusters is set to one or two, as the samples belong to the same category, which is crucial for classification applications. Therefore,

removing noisy samples should not split the samples of a category into more than two clusters. $X_{DBSCAN}^{(C_i)}$ expresses the samples of category $C_i$ (where $i = 1, \ldots, m$ and $C_i \neq C_{max}$). By concatenating the residual data from the reduced noise categories with the category that has the maximum samples, the data matrix is constructed as follows:

$$X_{DBSCAN} = \left[ X^{(C_{max})}, \bigcup_{\substack{i = 1 \\ i \neq max}}^{m} X_{DBSCAN}^{(C_i)} \right] \tag{3}$$

Finally, balanced data ($X_{CRN\text{-}SMOTE}$) is obtained using the SMOTE technique on $X_{DBSCAN}$ as follows:

$$X_{CRN-SMOTE} = \text{SMOTE}\left( K_{SMOTE}, \left[ X^{(C_{max})}, \bigcup_{\substack{i = 1 \\ i \neq max}}^{m} X_{DBSCAN}^{(C_i)} \right] \right) = \text{SMOTE}(K_{SMOTE}, X_{DBSCAN}) \tag{4}$$

In the proposed CRN-SMOTE, the cluster-based approach effectively reduces noise in the data by ensuring that samples from each category form one or two clusters. This is crucial as it helps prevent the misclassification of minority class samples that could occur due to noise in the dataset. Unlike many traditional data-level methods that merely aim to balance datasets without considering classification implications, CRN-SMOTE is specifically designed with classification performance in mind. This results in a more targeted approach to addressing imbalanced data issues. However, while the focus on clustering can enhance performance, it may also introduce implementation complexity. Additionally, the effectiveness of CRN-SMOTE depends on the quality of the clustering process; if the clustering parameters are not optimally set or if the underlying data structure is complex, it may lead to poor performance or misclassification.

## 4. Experiments and results

The proposed CRN-SMOTE method is evaluated across a range of imbalanced datasets from diverse domains. Performance comparisons between CRN-SMOTE and RN-SMOTE (the state-of-the-art method) are conducted using standard metrics for evaluating imbalanced data classification, including Kappa, Precision, Recall, F1-score, and MCC [52, 53]. Fig 5 illustrates the methodology used for evaluating the various datasets. For datasets with sufficient samples, the data is split into training and test sets, with the training dataset balanced using conventional methods to enhance classifier performance. Additionally, K-fold cross-validation is employed for datasets with fewer samples, where the training folds are balanced and fed into the classifier. The datasets and evaluation metrics are then detailed. The efficacy of the proposed method is assessed using three prominent classifiers—Support Vector Machine (SVM), Random Forest (RF), and AdaBoost (ADA)—utilizing their default parameters.

Furthermore, the results of the proposed CRN-SMOTE method are compared with two other state-of-the-art SMOTE-based methods: SMOTE-Tomek Link [54] and SMOTE-ENN [55] for further investigation.

### 4.1. Datasets

In this study, four conventional imbalanced datasets from the UCI Machine Learning Repository are utilized, and their specific details are provided in Table 1 [56]. These datasets are normalized to ensure that features with different scales did not bias the classifiers.

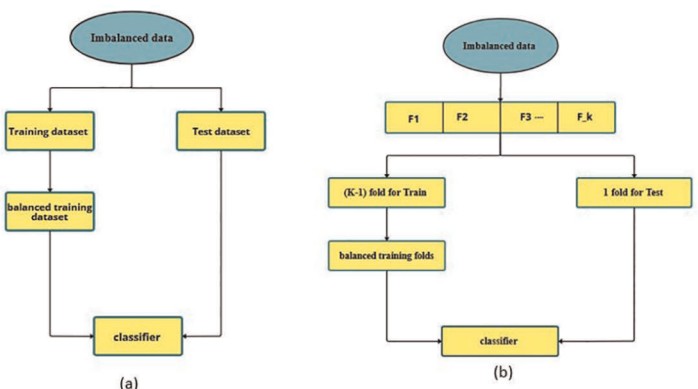

**Fig 5. The methodology for dividing the samples of datasets for evaluation in classification methods: (a) splitting the datasets into training and test sets for datasets with sufficient samples; (b) using k-fold cross-validation for evaluating datasets with fewer samples.**

**Table 1. The used datasets with their details.**

| Dataset | Total instance | Number of classes | Class1 | Class 2 | Class3 |
|---|---|---|---|---|---|
| QSAR | 1055 | 2 | 699 | 356 | - |
| Blood | 748 | 2 | 570 | 178 | - |
| ILPD | 583 | 2 | 416 | 167 | - |
| Maternal Health Risk | 1014 | 3 | 406 | 336 | 272 |

## 4.2. Evaluation metrics

One of the significant challenges in working with imbalanced datasets is selecting appropriate evaluation criteria. Due to the inherent imbalance in class distribution, the accuracy metric may not be sufficient for assessing the performance of classification models. In this study, we utilized a set of criteria that have proven effective in evaluating imbalanced datasets, including Kappa, MCC, F1 score, Precision, and Recall. These metrics are derived from the confusion matrix, a visual representation (such as Table 2) that summarizes the model's performance by comparing predicted labels to actual labels.

A confusion matrix is a valuable tool used to assess the performance of a machine learning model or classification algorithm by comparing predictions to actual true labels. Table 3 presents the formulas for common metrics derived from a confusion matrix.

In the kappa metric, $p_o$ represents the relative observed agreement among rates, while $p_e$ denotes the hypothetical probability of chance agreement. These parameters define as follows:

$$p_o = \frac{TN + TP}{TN + TP + FP + FN} \tag{5}$$

**Table 2. The definition of a confusion matrix.**

| | | Predicted | |
|---|---|---|---|
| | | Negative (-) | Positive (+) |
| Actual | Negative (-) | True negative (TN) | False Positive (FP) |
| | Positive (+) | False negative (FN) | True Positive (TP) |

**Table 3. Imbalanced classification metric.**

| metrics | precision | Recall | F1-Score | MCC | kappa |
|---|---|---|---|---|---|
| **definition** | $\frac{TP}{TP+FP}$ | $\frac{TP}{TP+FN}$ | $\frac{2*precision*recall}{precision+recall}$ | $\frac{TP*TN-FP*FN}{\sqrt{(TP+FP)(TP+FN)(TN+FN)}}$ | $\frac{p_o-p_e}{1-p_e}$ |

$$p_e = \frac{(TP+FP)(TP+FN)+(TN+FN)(TN+FP)}{(TP+TN+FP+FN)^2} \tag{6}$$

## 4.3. Simulation results

In the following, the results (the values of evaluation metrics) are presented for different settings on four mentioned datasets using Support Vector Machine (SVM), Random Forest (RF), and Ada Boost (ADA) classifiers. The results are calculated on the original imbalanced data, balanced data using the state-of-the-art RN-SMOTE method, and balanced data using the proposed CRN-SMOTE method with cluster sizes of 1 (1CRN-SMOTE) and 2 (2CRN-SMOTE). In all cases, the data is balanced using the SMOTE($K_{SMOTE}$, **X**) method, where $K_{SMOTE}$ is set to 4, 5, and 6. A 10-fold cross-validation methodology is applied to obtain and compare the results without any correlation to the dataset, and the average of the results are reported. Tables 4–6 show the results using SVM, ADA, and RF classifiers, respectively.

**Table 4. The average values of evaluation metrics on ILDP, QSAR, Blood and Health risk imbalanced datasets using SVM classifiers and 10-fold cross validation methodology.**

| dataset | The value of $K_{SMOTE}$ | $K_{SMOTE}=4$ | | | | | $K_{SMOTE}=5$ | | | | | $K_{SMOTE}=6$ | | | | |
|---|---|---|---|---|---|---|---|---|---|---|---|---|---|---|---|---|
| | metric | kappa | MCC | F1 | Pre | Recall | kappa | MCC | F1 | Pre | Recall | kappa | MCC | F1 | Pre | Recall |
| **ILPD** | Original Imbalanced data | 0.0 | 0.0 | 0.594 | 0.509 | 0.713 | 0.0 | 0.0 | 0.594 | 0.509 | 0.713 | 0.0 | 0.0 | 0.594 | 0.509 | 0.713 |
| | Balanced data using RN-SMOTE | 0.303 | 0.351 | 0.651 | 0.758 | 0.636 | 0.313 | 0.364 | 0.658 | 0.765 | 0.643 | 0.291 | 0.340 | 0.644 | 0.753 | 0.629 |
| | Balanced data using 1CRN-SMOTE | 0.308 | 0.357 | 0.653 | 0.761 | 0.637 | 0.315 | 0.366 | 0.659 | 0.765 | 0.644 | 0.296 | 0.3443 | 0.648 | 0.755 | 0.632 |
| | Balanced data using 2CRN-SMOTE | 0.319 | 0.369 | 0.661 | 0.766 | 0.646 | 0.319 | 0.370 | 0.663 | 0.767 | 0.648 | 0.322 | 0.371 | 0.665 | 0.767 | 0.649 |
| **QSAR** | Original Imbalanced data | 0.715 | 0.717 | 0.873 | 0.874 | 0.874 | 0.715 | 0.717 | 0.873 | 0.874 | 0.874 | 0.715 | 0.717 | 0.873 | 0.874 | 0.874 |
| | Balanced data using RN-SMOTE | 0.709 | 0.710 | 0.869 | 0.871 | 0.868 | 0.716 | 0.718 | 0.872 | 0.874 | 0.871 | 0.715 | 0.716 | 0.871 | 0.873 | 0.871 |
| | Balanced data using 1CRN-SMOTE | 0.719 | 0.720 | 0.873 | 0.875 | 0.873 | 0.729 | 0.730 | 0.877 | 0.88 | 0.876 | 0.728 | 0.729 | 0.877 | 0.879 | 0.876 |
| | Balanced data using 2CRN-SMOTE | 0.705 | 0.707 | 0.867 | 0.869 | 0.866 | 0.721 | 0.723 | 0.874 | 0.877 | 0.873 | 0.716 | 0.717 | 0.872 | 0.874 | 0.872 |
| **Blood** | Original Imbalanced data | 0.181 | 0.233 | 0.724 | 0.748 | 0.780 | 0.181 | 0.233 | 0.724 | 0.748 | 0.780 | 0.181 | 0.233 | 0.724 | 0.748 | 0.780 |
| | Balanced data using RN-SMOTE | 0.296 | 0.310 | 0.720 | 0.755 | 0.708 | 0.324 | 0.337 | 0.733 | 0.765 | 0.721 | 0.310 | 0.333 | 0.716 | 0.767 | 0.699 |
| | Balanced data using 1CRN-SMOTE | 0.323 | 0.331 | 0.738 | 0.760 | 0.729 | 0.397 | 0.406 | 0.771 | 0.788 | 0.765 | 0.343 | 0.362 | 0.734 | 0.776 | 0.720 |
| | Balanced data using 2CRN-SMOTE | 0.336 | 0.350 | 0.740 | 0.769 | 0.731 | 0.330 | 0.343 | 0.737 | 0.766 | 0.725 | 0.324 | 0.343 | 0.726 | 0.769 | 0.711 |
| **Health risk** | Original Imbalanced data | 0.544 | 0.560 | 0.687 | 0.706 | 0.704 | 0.544 | 0.560 | 0.687 | 0.706 | 0.704 | 0.544 | 0.560 | 0.687 | 0.706 | 0.704 |
| | Balanced data using RN-SMOTE | 0.541 | 0.555 | 0.684 | 0.699 | 0.701 | 0.524 | 0.538 | 0.671 | 0.685 | 0.690 | 0.531 | 0.542 | 0.681 | 0.694 | 0.694 |
| | Balanced data using 1CRN-SMOTE | 0.556 | 0.572 | 0.693 | 0.711 | 0.710 | 0.536 | 0.552 | 0.679 | 0.697 | 0.698 | 0.542 | 0.562 | 0.688 | 0.703 | 0.702 |
| | Balanced data using 2CRN-SMOTE | 0.508 | 0.540 | 0.637 | 0.626 | 0.679 | 0.470 | 0.511 | 0.582 | 0.557 | 0.653 | 0.465 | 0.518 | 0.558 | 0.510 | 0.650 |

**Table 5. The average values of evaluation metrics on ILDP, QSAR, Blood and Health risk imbalanced datasets using ADA classifiers and 10-fold cross validation methodology.**

| dataset | The value of $K_{SMOTE}$ | $K_{SMOTE}$ = 4 | | | | | $K_{SMOTE}$ = 5 | | | | | $K_{SMOTE}$ = 6 | | | | |
|---|---|---|---|---|---|---|---|---|---|---|---|---|---|---|---|---|
| | *metric* | kappa | MCC | F1 | Pre | Recall | kappa | MCC | F1 | Pre | Recall | kappa | MCC | F1 | Pre | Recall |
| *ILPD* | Original Imbalanced data | 0.231 | 0.238 | 0.692 | 0.692 | 0.704 | 0.231 | 0.238 | 0.692 | 0.692 | 0.704 | 0.231 | 0.238 | 0.692 | 0.692 | 0.704 |
| | Balanced data using RN-SMOTE | 0.276 | 0.290 | 0.676 | 0.716 | 0.663 | 0.236 | 0.248 | 0.663 | 0.698 | 0.651 | 0.230 | 0.244 | 0.658 | 0.697 | 0.646 |
| | Balanced data using 1CRN-SMOTE | 0.317 | 0.333 | 0.697 | 0.735 | 0.685 | 0.291 | 0.305 | 0.687 | 0.722 | 0.675 | 0.278 | 0.290 | 0.681 | 0.716 | 0.670 |
| | Balanced data using 2CRN-SMOTE | 0.348 | 0.362 | 0.713 | 0.747 | 0.703 | 0.331 | 0.344 | 0.707 | 0.738 | 0.696 | 0.342 | 0.357 | 0.718 | 0.744 | 0.711 |
| *QSAR* | Original Imbalanced data | 0.652 | 0.653 | 0.845 | 0.845 | 0.846 | 0.652 | 0.653 | 0.845 | 0.845 | 0.846 | 0.652 | 0.653 | 0.845 | 0.845 | 0.846 |
| | Balanced data using RN-SMOTE | 0.661 | 0.663 | 0.847 | 0.849 | 0.846 | 0.679 | 0.681 | 0.855 | 0.858 | 0.854 | 0.671 | 0.672 | 0.852 | 0.854 | 0.851 |
| | Balanced data using 1CRN-SMOTE | 0.718 | 0.718 | 0.873 | 0.874 | 0.873 | 0.720 | 0.720 | 0.874 | 0.875 | 0.874 | 0.722 | 0.723 | 0.875 | 0.877 | 0.874 |
| | Balanced data using 2CRN-SMOTE | 0.679 | 0.680 | 0.855 | 0.857 | 0.855 | 0.693 | 0.694 | 0.861 | 0.864 | 0.860 | 0.679 | 0.680 | 0.856 | 0.857 | 0.856 |
| *Blood* | Original Imbalanced data | 0.328 | 0.360 | 0.771 | 0.781 | 0.798 | 0.328 | 0.360 | 0.771 | 0.781 | 0.798 | 0.328 | 0.360 | 0.771 | 0.781 | 0.798 |
| | Balanced data using RN-SMOTE | 0.300 | 0.314 | 0.723 | 0.756 | 0.709 | 0.276 | 0.290 | 0.706 | 0.748 | 0.688 | 0.291 | 0.306 | 0.714 | 0.754 | 0.697 |
| | Balanced data using 1CRN-SMOTE | 0.316 | 0.327 | 0.730 | 0.759 | 0.715 | 0.322 | 0.331 | 0.735 | 0.761 | 0.722 | 0.312 | 0.326 | 0.725 | 0.761 | 0.710 |
| | Balanced data using 2CRN-SMOTE | 0.341 | 0.355 | 0.738 | 0.772 | 0.725 | 0.317 | 0.330 | 0.727 | 0.762 | 0.711 | 0.327 | 0.339 | 0.733 | 0.764 | 0.719 |
| *Health risk* | Original Imbalanced data | 0.484 | 0.487 | 0.663 | 0.674 | 0.662 | 0.484 | 0.487 | 0.663 | 0.674 | 0.662 | 0.484 | 0.487 | 0.663 | 0.674 | 0.662 |
| | Balanced data using RN-SMOTE | 0.393 | 0.405 | 0.599 | 0.636 | 0.603 | 0.394 | 0.404 | 0.603 | 0.639 | 0.604 | 0.473 | 0.479 | 0.656 | 0.671 | 0.653 |
| | Balanced data using 1CRN-SMOTE | 0.500 | 0.527 | 0.644 | 0.668 | 0.671 | 0.504 | 0.530 | 0.664 | 0.678 | 0.675 | 0.530 | 0.555 | 0.690 | 0.703 | 0.691 |
| | Balanced data using 2CRN-SMOTE | 0.467 | 0.520 | 0.614 | 0.622 | 0.651 | 0.495 | 0.531 | 0.609 | 0.570 | 0.670 | 0.442 | 0.516 | 0.537 | 0.487 | 0.636 |

The results presented in Tables 4–6 demonstrate the superior performance of the proposed cluster-based CRN-SMOTE method in comparison to the state-of-the-art RN-SMOTE. The results show that both 1CRN-SMOTE and 2CRN-SMOTE improve the classifiers' performance based on conventional evaluation metrics for imbalanced datasets in most settings.

For a more comprehensive investigation, the results of RN-SMOTE, 1CRN-SMOTE, and 2CRN-SMOTE on each dataset using all three classifiers (SVM, Random Forest, and Ada Boost) are given in Figs 6–9. The average of the metric values is reported for $K_{SMOTE}$ = 5 and using a 10-fold cross-validation evaluation.

The results again confirm the superior performance of the proposed CRN-SMOTE method compared to the RN-SMOTE approach. The cluster-based CRN-SMOTE techniques demonstrate improved classification performance across the various datasets and classifiers evaluated.

In continuation, the results of the proposed CRN-SMOTE (i.e., 1CRN-SMOTE) are compared with SMOTE-Tomek Link and SMOTE-ENN, two state-of-the-art SMOTE-based methods, as well as RN-SMOTE. The results are provided for $K_{SMOTE}$ = 5 and the RF classifier across all four mentioned datasets (see Tables 7 and 8). The findings demonstrate that the proposed CRN-SMOTE method outperforms other state-of-the-art methods in most cases. On the Blood dataset, SMOTE-ENN outperforms the proposed CRN-SMOTE method, while on all three other datasets, the proposed 1CRN-SMOTE demonstrates the best performance.

**Table 6. The average values of evaluation metrics on ILDP, QSAR, Blood and Health risk imbalanced datasets using RF classifiers and 10-fold cross validation methodology.**

| dataset | The value of $K_{SMOTE}$ | $K_{SMOTE} = 4$ | | | | | $K_{SMOTE} = 5$ | | | | | $K_{SMOTE} = 6$ | | | | |
|---|---|---|---|---|---|---|---|---|---|---|---|---|---|---|---|---|
| | *metric* | kappa | MCC | F1 | Pre | Recall | kappa | MCC | F1 | Pre | Recall | kappa | MCC | F1 | Pre | Recall |
| *ILPD* | Original Imbalanced data | 0.231 | 0.244 | 0.697 | 0.697 | 0.718 | 0.231 | 0.244 | 0.697 | 0.697 | 0.718 | 0.231 | 0.244 | 0.697 | 0.697 | 0.718 |
| | Balanced data using RN-SMOTE | 0.305 | 0.308 | 0.707 | 0.719 | 0.703 | 0.322 | 0.327 | 0.715 | 0.727 | 0.711 | 0.305 | 0.311 | 0.707 | 0.721 | 0.703 |
| | Balanced data using 1CRN-SMOTE | 0.360 | 0.364 | 0.732 | 0.742 | 0.728 | 0.352 | 0.358 | 0.727 | 0.740 | 0.723 | 0.333 | 0.340 | 0.719 | 0.733 | 0.716 |
| | Balanced data using 2CRN-SMOTE | 0.362 | 0.365 | 0.734 | 0.742 | 0.732 | 0.374 | 0.380 | 0.736 | 0.749 | 0.732 | 0.362 | 0.365 | 0.733 | 0.742 | 0.730 |
| *QSAR* | Original Imbalanced data | 0.722 | 0.725 | 0.877 | 0.878 | 0.879 | 0.722 | 0.725 | 0.877 | 0.878 | 0.879 | 0.722 | 0.725 | 0.877 | 0.878 | 0.879 |
| | Balanced data using RN-SMOTE | 0.717 | 0.719 | 0.874 | 0.875 | 0.875 | 0.719 | 0.722 | 0.876 | 0.876 | 0.877 | 0.702 | 0.704 | 0.868 | 0.868 | 0.869 |
| | Balanced data using 1CRN-SMOTE | 0.749 | 0.751 | 0.888 | 0.889 | 0.890 | 0.756 | 0.757 | 0.891 | 0.892 | 0.892 | 0.751 | 0.753 | 0.889 | 0.890 | 0.890 |
| | Balanced data using 2CRN-SMOTE | 0.728 | 0.729 | 0.879 | 0.879 | 0.880 | 0.726 | 0.728 | 0.878 | 0.879 | 0.880 | 0.732 | 0.735 | 0.881 | 0.883 | 0.883 |
| *Blood* | Original Imbalanced data | 0.192 | 0.198 | 0.719 | 0.712 | 0.736 | 0.192 | 0.198 | 0.719 | 0.712 | 0.736 | 0.192 | 0.198 | 0.719 | 0.712 | 0.736 |
| | Balanced data using RN-SMOTE | 0.208 | 0.211 | 0.715 | 0.714 | 0.719 | 0.182 | 0.184 | 0.703 | 0.705 | 0.707 | 0.193 | 0.195 | 0.707 | 0.709 | 0.708 |
| | Balanced data using 1CRN-SMOTE | 0.227 | 0.229 | 0.719 | 0.720 | 0.721 | 0.200 | 0.201 | 0.703 | 0.711 | 0.698 | 0.240 | 0.242 | 0.727 | 0.726 | 0.730 |
| | Balanced data using 2CRN-SMOTE | 0.255 | 0.258 | 0.730 | 0.732 | 0.732 | 0.234 | 0.235 | 0.719 | 0.723 | 0.719 | 0.239 | 0.242 | 0.728 | 0.726 | 0.733 |
| *Health risk* | Original Imbalanced data | 0.788 | 0.792 | 0.861 | 0.868 | 0.860 | 0.788 | 0.792 | 0.861 | 0.868 | 0.860 | 0.788 | 0.792 | 0.861 | 0.868 | 0.860 |
| | Balanced data using RN-SMOTE | 0.769 | 0.772 | 0.847 | 0.853 | 0.848 | 0.771 | 0.774 | 0.848 | 0.854 | 0.849 | 0.765 | 0.767 | 0.845 | 0.850 | 0.845 |
| | Balanced data using 1CRN-SMOTE | 0.798 | 0.800 | 0.867 | 0.871 | 0.867 | 0.792 | 0.795 | 0.862 | 0.868 | 0.862 | 0.795 | 0.798 | 0.865 | 0.871 | 0.864 |
| | Balanced data using 2CRN-SMOTE | 0.705 | 0.727 | 0.774 | 0.758 | 0.806 | 0.615 | 0.656 | 0.688 | 0.649 | 0.751 | 0.543 | 0.603 | 0.610 | 0.553 | 0.702 |

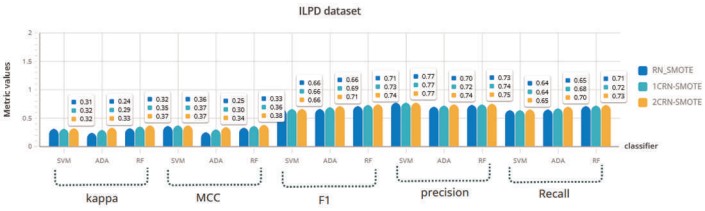

**Fig 6. Comparison of CRN_SMOTE results with RN-SMOTE in ILPD dataset for $K_{SMOTE} = 5$.**

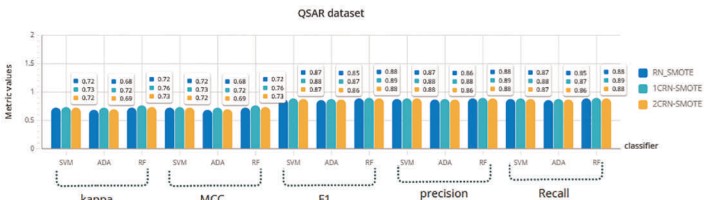

**Fig 7. Comparison of CRN_SMOTE results with RN-SMOTE on QSAR dataset for $K_{SMOTE} = 5$.**

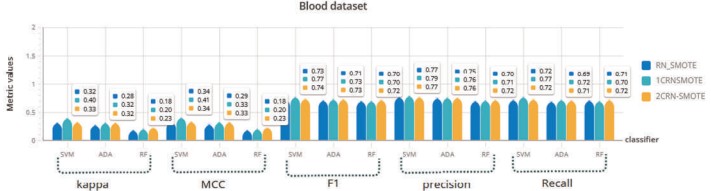

**Fig 8. Comparison of CRN_SMOTE results with RN_SMOTE on Blood dataset for $K_{SMOTE}$ = 5.**

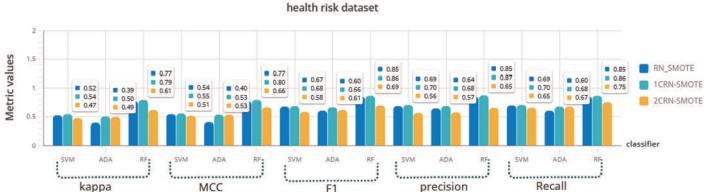

**Fig 9. Comparison of CRN_SMOTE results with RN_SMOTE on Health risk dataset for $K_{SMOTE}$ = 5.**

**Table 7. A comparison of the RN-SMOTE, SMOTE-Tomek Link, SMOTE-ENN, and the proposed 1CRN-SMOTE methods on the ILPD and QSAR datasets is presented, based on various classification metrics using the Random Forest classifier.**

| | *Dataset* | | | | | | | | | |
| | *ILPD* | | | | | *QSAR* | | | | |
| *method* | kappa | MCC | F1 | Pre | Recall | kappa | MCC | F1 | Pre | Recall |
| Proposed method (1CRN-SMOTE) | **0.352** | **0.358** | **0.727** | **0.740** | **0.723** | **0.756** | **0.757** | **0.891** | **0.892** | **0.892** |
| SMOTE-ENN [55] | 0.264 | 0.309 | 0.632 | 0.739 | 0.617 | 0.675 | 0.677 | 0.853 | 0.856 | 0.852 |
| SMOTE-TOMEK LINK [54] | 0.274 | 0.280 | 0.687 | 0.707 | 0.687 | 0.722 | 0.723 | 0.876 | 0.876 | 0.877 |
| RN-SMOTE [45] | 0.322 | 0.327 | 0.715 | 0.727 | 0.711 | 0.719 | 0.722 | 0.876 | 0.876 | 0.877 |

**Table 8. A comparison of the RN-SMOTE, SMOTE-Tomek Link, SMOTE-ENN, and the proposed 1CRN-SMOTE methods on the Blood and Health-risk datasets is presented, based on various classification metrics using the Random Forest classifier.**

| | *Dataset* | | | | | | | | | |
| | *Blood* | | | | | *Health-risk* | | | | |
| *method* | kappa | MCC | F1 | Pre | Recall | kappa | MCC | F1 | Pre | Recall |
| Proposed method (1CRN-SMOTE) | 0.200 | 0.201 | 0.703 | 0.711 | 0.698 | **0.792** | **0.795** | **0.862** | **0.868** | **0.862** |
| SMOTE-ENN [55] | **0.317** | **0.330** | **0.730** | **0.762** | **0.717** | 0.587 | 0.594 | 0.723 | 0.731 | 0.7297 |
| SMOTE-TOMEK LINK [54] | 0.196 | 0.197 | 0.702 | 0.709 | 0.700 | 0.772 | 0.775 | 0.862 | 0.855 | 0.850 |
| RN-SMOTE [45] | 0.182 | 0.184 | 0.703 | 0.705 | 0.707 | 0.771 | 0.774 | 0.848 | 0.854 | 0.849 |

## 4.4. Discussion

In this subsection, we compare the standard DBSCAN with the proposed method for removing noisy samples. The ε value significantly affects the performance of DBSCAN and must be adapted to the dataset's density. Determining the appropriate fraction value in the *K-distance*

algorithm is empirical and is typically based on the distances between data points. In classification applications, it is expected that samples belonging to the same category form a single cluster. However, using DBSCAN to remove noisy samples often results in multiple clusters for each category, which is unsuitable for classification tasks. This can negatively impact performance when addressing imbalanced data classification.

Fig 10 illustrates a case that exemplifies the challenges observed in the maternal health risk dataset [33], which consists of three distinct categories. The results indicate that DBSCAN divides the samples from minority classes into six clusters (Fig 10a and 10b), which may not be ideal for classification purposes. In contrast, our proposed method groups samples from each category into one or two clusters. The results presented in Table 9 demonstrate the effectiveness of our approach, as indicated by the classification metrics, which were obtained using the Random Forest classifier. In RN-SMOTE, the standard DBSCAN algorithm is applied.

## 5. Future work

A systematic study can be conducted to analyze how the oversampling process, facilitated by SMOTE, interacts with the denoising phase using DBSCAN. Understanding this relationship is crucial for identifying whether specific oversampling strategies yield better noise reduction outcomes or if particular noise characteristics influence the effectiveness of synthetic sample generation. For future work, we propose the following two suggestions:

1) Evaluating the Correlation between Oversampling and Noise Reduction: Investigating the correlation between the oversampling step and the noise reduction phase presents an attractive research direction that could significantly enhance the CRN-SMOTE method. This analysis could reveal insights into how these two processes influence each other and lead to more effective strategies for handling imbalanced datasets.

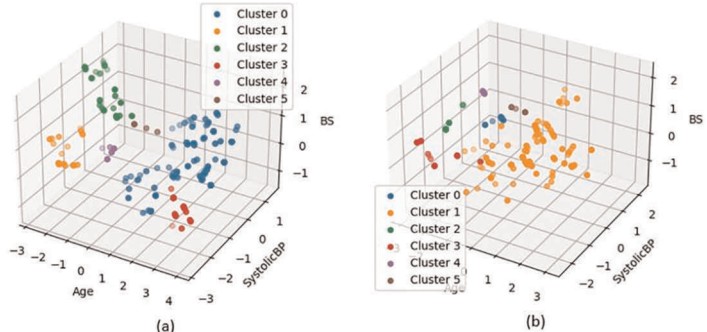

**Fig 10. Clusters generated using DBSCAN: (a) Clusters consisting of samples from category 1; (b) Clusters consisting of samples from category 2.**

**Table 9. A comparison of the CRN-SMOTE and RN-SMOTE methods on the health risk dataset based on different classification metrics using the Random Forest classifier.**

|  | kappa | mcc | F1 | pre | recall |
|---|---|---|---|---|---|
| **CRN-SMOTE** | **0.728** | **0.729** | **0.822** | **0.825** | **0.822** |
| **RN-SMOTE** | 0.668 | 0.670 | 0.782 | 0.785 | 0.783 |

2) Implementing a Mutual Neighborhood Check in SMOTE: We recommend incorporating a mutual neighborhood check within the SMOTE algorithm by utilizing a fixed number of neighbors. This approach aims to improve the quality of synthetic sample generation by ensuring that newly created samples are based not only on the closest minority class neighbors but also on their mutual relationships. By doing so, we can enhance the representativeness of the synthetic samples and mitigate the impact of noise.

## 6. Conclusion

The study presents a novel approach to addressing the prevalent issue of imbalanced data classification in machine learning through the introduction of Cluster-Based Reduced Noise SMOTE (CRN-SMOTE). This method effectively integrates the strengths of Synthetic Minority Over-sampling Technique (SMOTE) with a cluster-based noise reduction strategy, thereby enhancing the quality of synthetic samples generated for minority classes.

The results of the experiments conducted on four distinct imbalanced datasets demonstrate that CRN-SMOTE consistently outperforms the existing state-of-the-art methods, Reduced Noise SMOTE (RN-SMOTE), SMOTE-Tomek Link, and SMOTE-ENN, across all evaluation metrics, including Cohen's kappa, Matthew's correlation coefficient (MCC), F1-score, precision, and recall. The significant improvements observed—averaging 6.6% in Kappa and 4.01% in MCC—underscore the effectiveness of the proposed method in improving classification performance, particularly in challenging datasets such as QSAR and Maternal Health Risk.

In conclusion, the CRN-SMOTE method not only addresses the critical challenge of imbalanced data but also preserves the integrity of class distributions while significantly reducing noise. Its systematic approach to oversampling, noise reduction, and clustering control establishes a robust framework that can be applied across various domains and classification tasks. The empirical evidence presented validates its superiority over traditional methods, making it a valuable contribution to the field of machine learning and data classification.

## Supporting information

**S1 Datasets.**
(ZIP)

## Author Contributions

**Investigation:** Fakhroddin Nazari.

**Supervision:** Rassoul Hajizadeh.

**Validation:** Rassoul Hajizadeh.

**Writing – original draft:** Javad Hemmatian.

**Writing – review & editing:** Javad Hemmatian, Rassoul Hajizadeh, Fakhroddin Nazari.

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
