## [Decision Letter · Decision Letter 0]

17 Oct 2024

PONE-D-24-38797Addressing Imbalanced Data Classification with Cluster-Based Reduced Noise SMOTEPLOS ONE

Dear Dr. Hajizadeh,

Thank you for submitting your manuscript to PLOS ONE. After careful consideration, we feel that it has merit but does not fully meet PLOS ONE’s publication criteria as it currently stands. Therefore, we invite you to submit a revised version of the manuscript that addresses the points raised during the review process.

We look forward to receiving your revised manuscript.

Kind regards,

Shahid Akbar, PhD

Academic Editor

PLOS ONE

Journal Requirements:

2. Please note that PLOS ONE has specific guidelines on code sharing for submissions in which author-generated code underpins the findings in the manuscript. In these cases, we expect all author-generated code to be made available without restrictions upon publication of the work. 

Please review our guidelines at https://journals.plos.org/plosone/s/materials-and-software-sharing#loc-sharing-code and ensure that your code is shared in a way that follows best practice and facilitates reproducibility and reuse.

3. We note that your Data Availability Statement is currently as follows: 

“All relevant data are within the manuscript and its Supporting Information files.”

**Additional Editor Comments:**

Major Revision

Reviewers' comments:

Reviewer's Responses to Questions

**Comments to the Author**

1. Is the manuscript technically sound, and do the data support the conclusions?

Reviewer #1: Yes

Reviewer #2: Partly

2. Has the statistical analysis been performed appropriately and rigorously? 

Reviewer #1: Yes

Reviewer #2: Yes

3. Have the authors made all data underlying the findings in their manuscript fully available?

Reviewer #1: Yes

Reviewer #2: Yes

4. Is the manuscript presented in an intelligible fashion and written in standard English?

Reviewer #1: No

Reviewer #2: Yes

5. Review Comments to the Author

Reviewer #1: The authors have done well but still, some points need to be addressed which increases the readability and look of the manuscript better.

Here are some points

1- The author mentions which dataset results have better accuracy in the abstract.

2- The literature review has two subheadings current research and research gap.

3- The author needs to read the following manuscript and add some applications of Machine learning in different fields in the literature review ”Enhancing software defect prediction: a framework with improved feature selection and ensemble machine learning” “Impact of 3G and 4G technology performance on customer satisfaction in the telecommunication industry” Software defect prediction using an intelligent ensemble-based model.

4- Future work must be a section

5- The author needs to add a table before the conclusion in which results have been compared with some other studies

6- There must be a discussion section at the end

7- The author needs to add tables and figures in the Manuscript rather than down the manuscript so that there will be a better look and easier for the reader

8- What are the issues of the imbalanced dataset and how was it resolved

9- Write the advantages and disadvantages of the proposed method.

10- What are the disadvantages of the Synthetic Minority Oversampling Technique

11- The conclusion must be concise as abstract

Reviewer #2: Overall the paper seems interesting. The topic is valid and well organized. However, the following recommendations are needs to be addressed in order to improve the quality of the papers.

1. The abstract needs more improvement, the authors should mention the results and improvement than existing methods.

2.The problem statement and motivation of the paper needs to be clearly mention to provide a clear background to the readers.

2. The authors should provide the main contributions in points at the end of introduction section.

3. I appreciate the valuable use of machine learning in the model. For the readers concerns, i suggest adding/citing the recent computational models related machine learning and oversampling such as AIPs-DeepEnC-GA, StackedEnC-AOP, DeepAVPTPPred,

iAFPs-Mv-BiTCN, AIPs-SnTCN and Deepstacked-AVPs in order to provide useful information to the readers .

4. How the authors handle the overfitting and generalization of the proposed model.

5. The authors should clearly mention the future directions and real life applications of the proposed study.

6. What are the limitations of the proposed model.

6. PLOS authors have the option to publish the peer review history of their article (what does this mean?). If published, this will include your full peer review and any attached files.

Reviewer #1: **Yes: **Tehseen Mazhar

Reviewer #2: No

---

## [Author Response · Author response to Decision Letter 0]

28 Nov 2024

Dear Dr. Akbar and Reviewers,

Thank you for your valuable feedback and suggestions regarding our manuscript titled "Addressing Imbalanced Data Classification with Cluster-Based Reduced Noise SMOTE" (PONE-D-24-38797). We have carefully considered each comment and made the necessary revisions to improve the clarity and quality of our work. The changes can be tracked in the revised manuscript, and a separate Response to Reviewers sheet has also been prepared.

Summary of Revisions:

Abstract Improvement: We have revised the abstract to include specific results and improvements compared to existing methods, as suggested by Reviewer #2.

Problem Statement and Motivation: We have clarified the problem statement and motivation in the introduction to provide a stronger background for readers.

Main Contributions: The main contributions of our study are now clearly outlined in bullet points at the end of the introduction.

Literature Review Enhancements: We have included additional references and applications of machine learning, as recommended by the reviewers.

Future Work Section: A dedicated section for future work has been added to outline potential directions for further research.

Comparison Table: We have added a table comparing our results with other studies, enhancing the manuscript's comprehensiveness.

Discussion Section: A discussion section has been included to address the implications of our findings.

Addressing Overfitting: We have elaborated on how our method addresses overfitting and generalization issues.

Limitations and Advantages: The advantages and disadvantages of our proposed method has been added to the revised manuscript.

Concise Conclusion: The conclusion has been revised to be more concise, mirroring the abstract's style.

We believe these revisions have strengthened our manuscript and addressed the concerns raised during the review process. We appreciate the reviewers' insights and hope that our revised submission meets the publication criteria for PLOS ONE.

Thank you for considering our revised manuscript. 

Kind regards,

Rassoul Hajizaded (Corresponding Author)

---

## [Decision Letter · Decision Letter 1]

30 Dec 2024

Addressing Imbalanced Data Classification with Cluster-Based Reduced Noise SMOTE

PONE-D-24-38797R1

Dear Dr. Hajizadeh,

We’re pleased to inform you that your manuscript has been judged scientifically suitable for publication and will be formally accepted for publication once it meets all outstanding technical requirements.

Kind regards,

Agbotiname Lucky Imoize

Academic Editor

PLOS ONE

Additional Editor Comments (optional):

Accept.

Reviewers' comments:

Reviewer's Responses to Questions

**Comments to the Author**

1. If the authors have adequately addressed your comments raised in a previous round of review and you feel that this manuscript is now acceptable for publication, you may indicate that here to bypass the “Comments to the Author” section, enter your conflict of interest statement in the “Confidential to Editor” section, and submit your "Accept" recommendation.

Reviewer #2: All comments have been addressed

Reviewer #3: (No Response)

2. Is the manuscript technically sound, and do the data support the conclusions?

Reviewer #2: Yes

Reviewer #3: Yes

3. Has the statistical analysis been performed appropriately and rigorously? 

Reviewer #2: Yes

Reviewer #3: Yes

4. Have the authors made all data underlying the findings in their manuscript fully available?

Reviewer #2: Yes

Reviewer #3: Yes

5. Is the manuscript presented in an intelligible fashion and written in standard English?

Reviewer #2: Yes

Reviewer #3: Yes

6. Review Comments to the Author

Reviewer #2: The required concerns are successfully incorporated by the authors and now the paper quality have improved.i recommend to accept the paper.

Reviewer #3: The manuscript titled "Addressing Imbalanced Data Classification with Cluster-Based Reduced Noise SMOTE" addresses a critical issue in machine learning: handling imbalanced datasets. The authors propose a novel method, Cluster-Based Reduced Noise SMOTE (CRN-SMOTE), which enhances oversampling via noise reduction using clustering techniques. The study is evaluated using multiple performance metrics across four datasets, showing promising results compared to established methods such as RN-SMOTE, SMOTE-Tomek Link, and SMOTE-ENN.

Strengths:

The topic is timely and addresses a significant gap in machine learning classification, particularly with imbalanced datasets.The problem of classifier bias toward majority classes is well-articulated, providing a clear motivation for the study. The integration of clustering for noise reduction within SMOTE is a novel approach, offering theoretical and practical contributions. The study utilizes multiple datasets (ILPD, QSAR, Blood, and Maternal Health Risk) and evaluates performance with a comprehensive set of metrics (Cohen's kappa, MCC, F1-score, precision, and recall). Consistent performance improvements across datasets demonstrate robustness and applicability. The results indicate a statistically significant improvement over state-of-the-art methods, which is well-supported by comparative data. The abstract and introduction effectively contextualize the study, making the research accessible to both academic and applied audiences. By advancing oversampling techniques, the manuscript contributes to improved classification performance for minority classes, particularly in critical domains such as health risk assessment. The implications extend beyond the datasets studied, offering potential benefits across various imbalanced datasets. Addressing class imbalance remains a critical research area in machine learning, with applications spanning healthcare, finance, and beyond. This manuscript aligns well with current trends and offers solutions that practitioners and researchers can readily adopt.

Conclusion

The manuscript provides a significant advancement in addressing imbalanced data classification. The proposed CRN-SMOTE method is well-motivated, thoroughly evaluated, and impactful, with the potential to influence both research and practice. The authors have demonstrated rigorous methodology, solid experimental design, and clear presentation of results. Therefore, I strongly recommend accepting the manuscript for publication

7. PLOS authors have the option to publish the peer review history of their article (what does this mean?). If published, this will include your full peer review and any attached files.

Reviewer #2: No

Reviewer #3: **Yes: **Richard Govada Joshua

---

## [Editor Report · Acceptance letter]

13 Jan 2025

PONE-D-24-38797R1 

PLOS ONE

Dear Dr. Hajizadeh, 

I'm pleased to inform you that your manuscript has been deemed suitable for publication in PLOS ONE. Congratulations! Your manuscript is now being handed over to our production team.

Kind regards, 

on behalf of

Mr. Agbotiname Lucky Imoize 

Academic Editor

PLOS ONE